# Municipal Landfill Leachate Treatment and Sustainable Ethanol Production: A Biogreen Technology Approach

**DOI:** 10.3390/microorganisms10050880

**Published:** 2022-04-22

**Authors:** Mahmod Sidati Ali Abobaker, Husnul Azan Tajarudin, Abdul Latif Ahmad, Wan Maznah Wan Omar, Charles Ng Wai Chun

**Affiliations:** 1School of Industrial Technology, Bioprocess Technology Division, Universiti Sains Malaysia, Penang 11800, Malaysia; mahmod85@student.usm.my (M.S.A.A.); charlesngwaichun@student.usm.my (C.N.W.C.); 2School of Chemical Engineering, Engineering Campus, Universiti Sains Malaysia, Nibong Tebal 14300, Malaysia; chlatif@usm.my; 3School of Biological Sciences, Universiti Sains Malaysia, Penang 11800, Malaysia; wmaznah@usm.my

**Keywords:** leachate treatment, *Scenedesmus* sp., *Saccharomyces cerevisiae*, ethanol, biomass, bioreactor

## Abstract

Sustainable material sources are an important agenda to protect the environment and to meet human needs. In this study, *Scenedesmus* sp. was used to treat municipal landfill leachate via batch and continuous cultivation modes to protect the environment and explore sufficient biomass production for bioethanol production using *Saccharomyces cerevisiae*. Physicochemical characteristics of leachate were determined for the phases before, during, and after the process. Batch and continuous cultivation were used to treat raw leachate to determine optimum conditions for treatment. Then, the biomass of *Scenedesmus* sp. with and without sonication was used as a substrate for ethanol production. Sonication was carried out for biomass cell disruption for 20 min at a frequency of 40 kHz. Through batch cultivation mode, it was found that pH 7 was the optimum condition for leachate treatment. Continuous cultivation mode had the highest removal values for COD, phosphorus, and carbohydrate, namely 82.81%, 79.70%, and 84.35%, respectively, among other modes. As for ethanol production, biomass without sonication with 9.026 mg·L^−1^ ethanol, a biomass concentration of 3.300 µg·L^−1^, and pH 5 were higher than biomass with sonication with 5.562 mg·L^−1^ ethanol, a biomass concentration of 0.110 µg·L^−1^, and pH 5. Therefore, it is evident that the leachate has the potential to be treated by *Scenedesmus* sp. and converted to bioethanol in line with the concept of sustainable materials.

## 1. Introduction

Landfilling has become the most common and vital means of municipal solid waste disposal globally (Wang et al., 2018). Landfill functions as a huge anaerobic bioreactor, covering a wide range of chemical, physical, and biological systems [1]. For decades, landfills have posed problems for most countries owing to the generation of leachate, which threatens groundwater and surface water, as well as the environment and human health. Leachate is a liquid extracted from the decomposition of solid waste that contains a myriad of organic and inorganic compounds [2], from the merged action of rainwater and natural fermentation of the secreted waste [3]. It may contain heavy metals, organic contaminants, mineral salts, and nitrogen composites, and it can also be identified according to various parameters, such as pH, total dissolved solids (TDS), heavy metals, biochemical oxygen demand (BOD), and chemical oxygen demand (COD) [4]. The leachate composition differs from one landfill to another, and this composition is determined by factors such as the age of the landfill and the type of waste [5,6]. Based on the age of the landfill, leachate can also be grouped as old, medium, and young. 

In recent years, more leachate treatment technologies have been developed in several countries, which can be grouped into chemical, physical, and biological treatments, and are highly dependent on the leachate physicochemical conditions to determine the most appropriate treatment to adopt [7]. In Malaysia, there is a lack of efficient and long-term integrated methods for leachate treatment, whereas traditional biological treatments are expensive, require a long operating time, and have a vast carbon footprint [8]. Biological treatments have been effective in treating young leachates containing high organic content and producing valuable products such as biogas and fertilizer [9]. This type of treatment involves the utilization of certain microorganisms to enhance and improve the leachate treatment capability [10]. As a result, algae-based treatment for leachate is an alternative to conventional biological treatments.

Owing to its numerous advantages, leachate treatment using algae has generated great interest among researchers in terms of its high energy production capacity, particularly biogas, in addition to its high growth rate and ease of cultivation. Compared to terrestrial raw materials, algae can grow 5 to 10 times in favorable conditions and also has higher production rates [11]. In addition, it is low-cost as algae can be found everywhere and can tolerate adverse environments [12]. Algae can grow in arid regions such as deserts or coastal plains (which are much less or even unable to produce food) and utilize nutrient-rich wastewater for their growth [13]. Furthermore, it is easy to convert microalgae to monosaccharides for ethanol production since they do not consist of lignin [14]. While treating leachate, the use of algae fulfills a dual function of degrading pollutants and producing useful bioproducts simultaneously [15]. These features make algae a promising medium for ethanol production and leachate treatment. 

One of the most promising algal species used as feedstock for ethanol production is *Scenedesmus* sp. Among some microalgae, *Scenedesmus* sp. can achieve the maximum biomass concentration even though it is cultivated under heterotrophic conditions, as reported by [16]. A heterotrophic system favors higher productivities and biomass concentrations throughout the year, at a large scale, which is not restricted by light penetration or local weather conditions [17]. *Scenedesmus* sp. also possesses beneficial features such as rapid growth, CO_2_ fixation, and the ability to grow in wastewaters and accumulate lipids, which act as the basis for its selection in microalgae studies. Moreover, yeast is another microorganism beneficial for assisting ethanol production from algae-based leachate treatment. Yeast is known to have the capability of removing organic matter from municipal wastewater, eliminating additional nutrient removal steps, and reducing treatment costs [17]. It also plays a crucial role as a fermenting agent to convert sugars to ethanol.

Several studies have reported on the use of combined microalgae and yeast cultivation in treating wastewater and producing valuable products. For example, this approach has been used for a pilot-scale urban wastewater treatment and biodiesel production [18], yeast industry liquid digestate treatment and biofuel production [19], synthetic wastewater and ethanol production [20], and municipal wastewater and bioethanol production [17] with a mixture of various microalgae and yeast strains. Nevertheless, the application of combined microalgae cultivation and yeast fermentation for improved leachate treatment and ethanol production has not been widely researched. To ensure the process’s sustainability, valuable products, such as ethanol, can be extracted along with the recovery and reduction of waste generation in the process. Thus, this study employed *Scenedesmus* sp. for landfill leachate treatment and, subsequently, the treated leachate was then used as a substrate for fermentation by yeast, namely *Saccharomyces*
*cerevisiae*, for ethanol production. This study also examined the effect of two cultivation modes, namely batch and continuous, for raw leachate treatment. This study aimed to find a sustainable treatment for landfill leachate that can contribute to energy research. The combination of the use of algae and yeast with leachate as a substrate could indicate the potential of producing ethanol in a sustainable manner from renewable energy sources.

## 2. Material and Methods

### 2.1. Sample Collection

In this study, leachate was collected from Ampang Jajar Municipal Solid Waste (MSW) Collection Station, Permatang Pauh, Penang (5.1437° N, 100.5012° E). The landfill has been in operation since 2005; it receives around 500 tons of municipal solid wastes daily, including 68% of organic materials, and the volume of leachate generated in this landfill is estimated to be approximately 270 m^3^ per day. The characteristics of the Ampang Jajar landfill site are as follows: 2.9 hectares, clay as lining material, 800 kg/m^3^ density of compacted waste, 2.3–2.5 m/year of average rainfall, and 1.3–1.4 m/year of evaporation [21]. Approximately 30 L of leachate was collected in July 2020 and stored in three containers made of refractory material, each with a capacity of 10 L. A volume of 200 mL of sample was taken for initial physicochemical analysis of leachate content. The remaining leachate waste was sterilized in an autoclave at 121 °C for 20 min, stored in a sterile environment, and used at varying times. Table 1 shows the characteristics of the raw leachate used in this research.

### 2.2. Algae Strain and Preparation of Cultivation

In this study, *Scenedesmus* sp. was collected from a river in Johor, Malaysia. The algae samples were collected using a 25 μm planktonic net and then stored in sterilized glass bottles until being transferred to the laboratory via a portable refrigeration box, before they were subjected to isolation. In the current study, *Scenedesmus* sp. was isolated using BBM medium containing 1.5% agar. Spread plates were inoculated with 0.1–0.2 mL of water sample and incubated for 5–7 days at 25 °C, 60 µmol photons m^−2^ s^−1^, and a 14:10 h light:dark photoperiod [22]. To prevent contamination, the top of the glass bottle was covered with a sterile cotton wool plug and organic-free water was used. The preparation of the algae was conducted through Bold’s Basal Medium (BBM), as described by [2]. 

### 2.3. Physicochemical Analysis of Leachate

After sampling, the leachate samples were analyzed for pH, chemical oxygen demand (COD), carbohydrate content, nitrogen, and phosphorus content. Analysis of pH was conducted immediately using a pH meter at the laboratory. The COD analysis, phosphorus content, and total Kjeldahl nitrogen (TKN) were conducted following the standard method of APHA (2005). For COD analysis, the reading was taken with a wavelength of 620 nm. For total Kjeldahl nitrogen (TKN) analysis, the samples were digested with sulfuric acid at 380 °C. The digested sample was then added with 10 mL of distilled water and the solution was distilled into 25 mL of 4% boric acid solution containing 5 drops of methyl red indicator with 35% sodium hydroxide using a steam distillation unit (Model: K-355, BUCHI, Zurich, Switzerland). The boric acid receiving solution was titrated with 0.02 M of hydrochloric acid solution. The volume of hydrochloric acid required for the solution to turn pink or purple was recorded. 

#### Determination of Carbohydrate Content

Approximately 200 mg of Anthrone reagent was dissolved in 100 mL of H_2_SO_4_, previously prepared by mixing 500 mL of concentrated acid with 200 mL of water, and cooled in the refrigerator before mixing. A standard glucose solution was prepared by adding 100 mg of glucose to 100 mL of distilled water. Then, 10 mL of this stock was diluted with 100 mL of distilled water to produce a working standard. This standard was used to prepare standard tubes with the following concentrations of 0, 0.2, 0.4, 0.6, 0.8, and 1 mL, with ‘0’ acting as the blank. All volumes were made up to 1 mL with distilled water if necessary. Two mL of leachate was then hydrolyzed in a boiling tube by it placing in a boiling water bath for 3 h after adding 5 mL of 2.5N HCL. Thereafter, the tubes were allowed to cool and then neutralized with solid sodium carbonate until the effervescence ceased. The samples were then made up to 100 mL and centrifuged. Approximately 1 mL aliquots of supernatant were taken for further analysis. Four mL of Anthrone reagent was added to all tubes (standards and samples) and heated for 8 min in a boiling water bath. The tubes were cooled rapidly, and the absorbance was read at a wavelength of 630 nm using a DR2800 spectrophotometer. 

### 2.4. Treatment of Raw Leachate Using Scenedesmus *sp.* in Batch and Continuous Bioreactors

In this experiment, batch and continuous modes were used. The geometry and size of the bioreactors were the same. The glass vessel was connected with some tubes, one of them for aeration and others for sampling in a batch bioreactor. In a continuous bioreactor, with a 800 mL level, there is media output to remove the excess media from this level; it is also connected with a peristaltic pump for the input of raw leachate with different dilution rates, as shown in Figure 1. The bioreactors were sterilized and the pump for the continuous system was calibrated. The working volume for batch and continuous modes was 800 mL. The flow rates for fresh leachate feed into the bioreactor for the continuous system were 120, 150, and 180 mL·h^−1^. The aeration flow rate was 0.5 mL/min and the temperature inside the bioreactors was 28 ± 2 °C. Since *Scenedesmus* sp. can grow mixotrophically and heterotrophically and grows well under light conditions, it tends to divide preferably under dark conditions by binary or multiple fission to produce daughter cells, and this seems to have a significant implication for the overall productivities of microalgal cultures. The duration and intensity of light directly affect the growth and photosynthesis of microalgae.

In this study, leachate was used as the culture medium for the cultivation of *Scenedesmus* sp. In the batch bioreactor, sterile leachate was filtered and transferred into four bioreactors, each with a volume of 800 mL. Each bioreactor was set at different pH, namely 6.5, 7, 7.5, and 8. The pH of the leachate was adjusted using H_2_SO_4_ (1 mol) and NaOH (1 mol). The inoculum was prepared before being transferred to the leachate; the strain (*Scenedesmus* sp.) was inoculated in BBM media at an initial pH of 7.4 ± 0.1 and grown photo-autotrophically at 28 ± 2 °C, with a 12/12 h photoperiod (light/dark cycle) and 60 µmol photons m^−2^ s^−1^ irradiance given by two 40-watt daylight fluorescent lights. Then, 10% of inoculum was prepared to be transferred to 800 mL of leachate. Algae cultivation was monitored for 15 days during all phases A is the lag phase, B is the exponential phase, C is the stationary phase, and D is the death phase, as shown in Figure 2 with daily OD readings recorded using the DR 2800 spectrophotometer at a wavelength of 680 nm. At the end of the 15 days, as shown in Figure 3, the treated leachate was then used for further analysis.

For the continuous bioreactor, the bioreactor was run for 33 days. pH 7 was set for the continuous bioreactor. After 15 days, fresh leachate was added to the bioreactor at intervals, and reactions were observed at varying flow rates to compare various treatment results. The first flow rate was 2 mL/min and ran from day 15 to day 18. The second flow rate was 2.5 mL/min, which was from day 19 to day 22. Meanwhile, the third flow rate was 3 mL/minute from day 29 to day 33. The growth of biomass was divided into four phases: A is the lag phase, B is the exponential phase, C is the stationary phase, and D is the death phase, as shown in Figure 3. The OD readings were recorded daily using a DR 2800 spectrophotometer at a wavelength of 680 nm in the culture room, as shown in Figure 4.

### 2.5. Preparation of Biomass 

The treated leachate was centrifuged at 4500 rpm for 15 min and then separated into two parts, namely supernatant and biomass. The biomass was then further divided into two parts: one part was subjected to sonication while the other was without sonication. Sonication was conducted for 20 min at a frequency of 40 kHz. Biomass samples were freeze-dried at −47 °C for 48 h. After that, the biomass was weighed (0.165 g) and divided into five equal parts to obtain different concentrations. These parts were then added to different volumes of water (100, 150, 200, 250, 300 mL) to form different biomass concentrations for use in ethanol production.

### 2.6. Conversion of Biomass to Ethanol 

Fermentation was conducted in anaerobic conditions for the conversion of biomass to ethanol. Biomass algae without and with sonication were placed in sterile storage vessels in a laminar flow cabinet in sterile conditions to prevent contamination. Each biomass group (biomass algae without and with sonication) was prepared in five different concentrations (3.300, 2.200, 0.165, 0.132, and 0.110 µg·L^−1^) to determine the most appropriate concentration for *S. cerevisiae* growth and subsequently ethanol production. Each sample dilution was then prepared in three different pH values, namely 5.0 (acidic), 6.5 (neutral), and 9.0 (alkaline), to determine the optimum pH conditions for *S. cerevisiae* growth and ethanol production. 

Before inoculation, the medium was injected with nitrogen gas using a gauge needle. The fermentation vials used were autoclaved at 121 °C for 15 min. The standard inoculum (YEPG) was prepared from *S. cerevisiae* cultured in a medium consisting of 1% yeast, 2% peptone, and 2% glucose, and mixed with 100 mL of distilled water for 19 h to obtain the optimized absorbance of approximately 1.2. Then, the growth medium was inoculated with 10% (*v*/*v*) *S. cerevisiae* culture. The fermentation process was conducted in the incubator at 30 °C for 24 h. Sampling was conducted for 24 h. The fermentation vials were taken out from the incubator and the OD readings were measured immediately. The concentration of ethanol produced in each sample was measured using gas chromatography (Agilent 5820-A) with a split/splitless inlet with a temperature of 290°, a flame ionization detector (FID), and a capillary column (HP-Innowax 30 m, 0.32 mm, 0.15 µm).

## 3. Results and Discussion

### 3.1. Physicochemical Characteristics of Raw Leachate

From the physicochemical analysis, it was found that the raw leachate was in an acidic condition at pH 5.25 containing 6583.00 mg·L^−1^ COD, 4.350 mg·L^−1^ phosphorus, 245.00 mg·L^−1^ nitrogen, and 0.812 mg·L^−1^ carbohydrates. The raw leachate in this study can be considered as young leachate as it indicated high COD (>5000 mg·L^−1^), low nitrogen content (<400 mg·L^−1^), and low pH (Rahman et al., 2019). A high COD value was recorded, which could be due to the high pollutant levels and the humic acid substrates present, which are unable to be stabilized by the microorganism. 

In general, leachate composition is dependent on the landfill nature, waste composition, and age [23].

### 3.2. Comparison of Leachate Treated with Scenedesmus *sp.* Using Continuous and Batch Modes 

The pH values of four leachate samples were regulated in batch mode for 15 days. The removal of COD, phosphorus, carbohydrate, and nitrogen was monitored. Figure 5 shows the percentage removal of these parameters and biomass production. All parameters exhibited optimum values at pH 7, indicating that this was the best condition for leachate treatment with *Scenedesmus* sp. Optimum pH will intensify the growth of biomass production and the removal of carbohydrates [24]. Thus, a range between pH 7 and 8 was considered the best condition for *Scenedesmus* sp. growth and induced the best removal percentage for COD, carbohydrate, and other parameters.

Nevertheless, although pH 7 exhibited the best values for physicochemical parameters, it recorded the lowest biomass of 0.560 g·L^−1^. In contrast, pH 8 had the highest biomass of 0.669 g·L^−1^. The composition of algal biomass is a vital indicator in assessing the potential of algae as a biofuel source [25]. pH affects the metabolic process of the algae, which includes its nutrient consumption. Unfavorable pH will restrict access to carbon, thus ceasing the formation of CO_2_. This inhibits cell growth and leads to low biomass/growth rates [26].

For continuous cultivation, three different flow rates were examined at pH 7. The flow rate, which is the speed of the support media in comparison to the culture broth, affects the algal growth [27]. Continuous treatment mode typically improves the performance of the entire process. It allows for increased productivity and tight control, minimizes uploading and downloading time, and reduces contamination risk [28]. Figure 6 shows the percentage removal of the parameters and biomass production under continuous mode. At a dilution rate of 2 mL/min, biomass production was the highest, at 0.674 g·L^−1^, despite demonstrating the lowest COD, carbohydrate, phosphorus, and nitrogen removal. In contrast, the dilution rate of 3 mL/min had the lowest biomass of 0.498 g·L^−1^ but had the highest COD, carbohydrate, phosphorus, and nitrogen removal. The optimum dilution rate will provide optimum biomass and percentage consumption or removal of carbohydrates [29].

Continuous mode for algae cultivation demonstrated higher biomass production compared to batch mode. Moreover, continuous mode treatment also indicated higher COD, carbohydrate, phosphorus, and nitrogen removal compared to batch mode. Batch reactor treatment is more difficult to apply for large-scale industrial purposes. The continuous mode is considered to be a less expensive option, capable of producing better results compared to the batch cultivation mode [30]. Similar to our findings, the authors in [29] also demonstrated excellent results when comparing the continuous and batch systems for microalgae conversion based on various parameters.

### 3.3. Harvesting and Conversion of Biomass for Ethanol Production

*S. cerevisiae* was selected due to its potential to produce a high ethanol yield (Aziz et al., 2020). Different pH conditions and biomass concentrations were used to show their influence on the growth of *S. cerevisiae* and ethanol production. pH is one of the factors that affects the nutrient uptake of an organism, cell aggregation, and cell membrane destabilization (Debnath et al., 2021). For this reason, optimum pH should be maintained in algal growth media to achieve maximum biomass and carbohydrate production [31].

The results of this study are similar to those of Gohain et al. (2021), who investigated the quantitative conversion of algal biomass to ethanol via fermentation by *S. cerevisiae.* The study found that *S. cerevisiae* was able to convert algal biomass to ethanol efficiently, with results that encourage the use of algal biomass as feedstock for ethanol production. In addition, the authors in [32] compared the use of various treatment modes and achieved 8.20 g·L^−1^. They concluded that the variation in bioethanol yield and concentration was due to the pre-treatment steps used. Similarly, they also obtained a better bioethanol yield and concentration as compared to previous studies. Microorganisms have many nutrient-rich elements in their bodies that can be utilized for other valuable products or chemicals.

#### 3.3.1. Removal of Nitrogen, Phosphorus, and Carbohydrate 

Nitrogen, phosphorous, and carbohydrate are the primary products of photosynthesis and thus store metabolic energy temporarily. The highest carbohydrate removal of 46% was obtained in this study. On another note, the authors in [33] also achieved maximum carbohydrate removal of 49.71%. Meanwhile, the highest removal of phosphorus and nitrogen was 28% and 11%, and these results correspond to the values reported by [34]. The relationships between biomass and each parameter are depicted in Figure 7 and Figure 8.

Based on the results obtained, sonication was found to increase the concentration of dissolved nutrients such as carbohydrates, phosphorus, and nitrogen. Sonication could be the main reason for decreasing nutrient removal through fermentation [35]. For this study, various biomass concentrations were subjected to fermentation via *S. cerevisiae* under sonication and without sonication. From Figure 7 and Figure 8, it was observed that sonication did not improve the removal of any of the parameters. On the contrary, fermentation has been successful in augmenting the removal of phosphorus and other biological nutrients such as nitrogen [26]. Furthermore, previous research has indicated that the rate of phosphorus removal has a positive correlation with pH ranging between 6.5 and 8.5 [36]. This is in line with this study, where the highest phosphorus removal was recorded at pH 9 after fermentation. In the fermentation of biomass without sonication, the highest phosphorus removal was observed at pH 5.

For phosphorus and carbohydrate, samples without sonication had higher removal than samples with sonication. At pH 9, phosphorus removal was the highest at 3.30 µg·L^−1^, while carbohydrate removal was the highest at 0.165 µg·L^−1^. At pH 6.5, carbohydrate removal was the highest at 0.110 µg·L^−1^. For biomass with sonication, only concentrations of 3.30 µg·L^−1^ and 2.20 µg·L^−1^ indicated slight nitrogen removal of 4.84% and 3.56%, respectively, at pH 5 and 9. Meanwhile, for biomass without sonication, higher nitrogen removal was recorded at 3.30 µg·L^−1^, 2.20 µg·L^−1^, and 0.165 µg·L^−1^. The values of nitrogen removal were 11.01%, 7.68%, 6.04%, 6.90%, and 8.21% at different pH values.

#### 3.3.2. Ethanol Production

Leachate is rich in a variety of nutrients that could be utilized to produce other valuable chemicals. Treated leachate is also a promising medium for bioethanol production. The continuous mode at pH 7 indicated optimum biomass and the removal of other undesired nutrients. In this study, fermentation was performed at three different pH values (5, 6.5, 9), and with and without sonication. The results for ethanol production are depicted in Figure 9 and Figure 10 are in line with the results from previous research [36]. The highest ethanol concentration of 9.020 mg L^−1^ was found in a biomass concentration of 3.300 µg·L^−1^ at pH 5 when fermentation occurred in samples without sonication. Microalgae, such as *Scenedesmus* sp. used in this study, are known to be extremely useful in the production of bioethanol. For fermentation, *S. cerevisiae* is considered one of the most effective yeasts to serve this purpose. It induces high bioethanol yields and can resist inhibitory components that may interfere with the fermentation process, resulting in high bioethanol concentrations. The optimum pH range for *S. cerevisiae* is 3.5 to 5 [37] or 4 to 5 [38]. This is supported by the fact that, in this study, the highest ethanol production was recorded at pH 5.

Figure 9 and Figure 10 illustrate the ethanol concentrations from various algae biomass concentrations and pH values. The results revealed that the biomass concentration of 3.30 µg·L^−1^ without sonication at pH 5 showed the highest ethanol concentration of 9.020 mg·L^−1^. The biomass concentration of 0.110 µg·L^−1^ at pH 6.5 obtained the lowest ethanol concentration of 1.19 mg·L^−1^. In contrast, the highest ethanol concentration of 5.56 mg·L^−1^ was obtained by the 0.11 µg·L^−1^ biomass concentration with sonication at pH 5, while, at a similar pH, the lowest ethanol concentration of 0.17 mg·L^−1^ was obtained at the 3.300 µg·L^−1^ biomass concentration. These results suggested that higher biomass concentrations without sonication favored higher ethanol concentrations, while lower biomass concentrations with sonication achieved higher ethanol concentrations. This was because *S. cerevisiae* will produce high ethanol concentrations with less or no oxygen (anaerobic) [32]. The sonication process allows oxygen to mix in the substrate and will significantly increase the dissolved oxygen, and eventually, an aerobic process will occur. During the process of sonication, a certain amount of oxygen is trapped in the biomass of algae. Then, at the same time, yeast degrades the algae, and obtains oxygen for algae. It supports the aerobic process (the process in present oxygen). This is why the amount of biomass is high but ethanol is low. However, without sonication, less oxygen is trapped in the biomass of algae, and this results in the anaerobic condition (without oxygen) and causes ethanol production to be high. Table 2 shows a comparison of results for ethanol concentrations at different biomass concentrations, with and without sonication, as well as pH values and maximum biomass of *S. cerevisiae*.

## 4. Conclusions

Algae demonstrates great potential for the production of various bioproducts such as bioethanol, biodiesel, and chemicals. According to the physicochemical characteristics of raw leachate, it recorded pH 5.25, 6583.30 mg·L^−1^ COD, 4.35 mg·L^−1^ phosphorus, 245.00 mg·L^−1^ nitrogen, and 0.812 mg·L^−1^ carbohydrates. For batch cultivation, pH 7 was the optimum condition for leachate treatment. Continuous cultivation indicated the highest removal of COD (82.81%), phosphorus (79.70%), and carbohydrate (84.35%). For ethanol production, the highest ethanol concentration of 9.020 mg·L^−1^ was obtained from a biomass concentration of 3.300 µg·L^−1^ at pH 5 when fermentation occurred in samples without sonication. Under sonication, the biomass concentration of 3.30 µg·L^−1^ at pH 5 exhibited the highest ethanol concentration of 9.020 mg·L^−1^. Based on the results obtained, the combined use of *Scenedesmus* sp. and *S. cerevisiae* for leachate treatment and ethanol production is promising to be established as a sustainable approach in maximizing waste management.

## Figures and Tables

**Figure 1 microorganisms-10-00880-f001:**
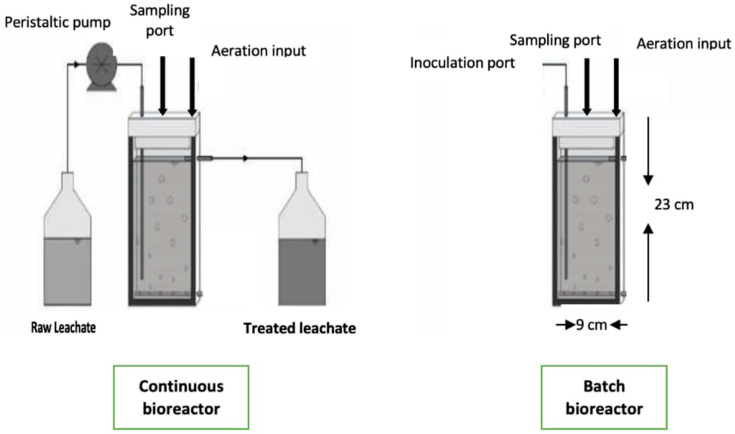
Batch and continuous bioreactors used for raw leachate treatment.

**Figure 2 microorganisms-10-00880-f002:**
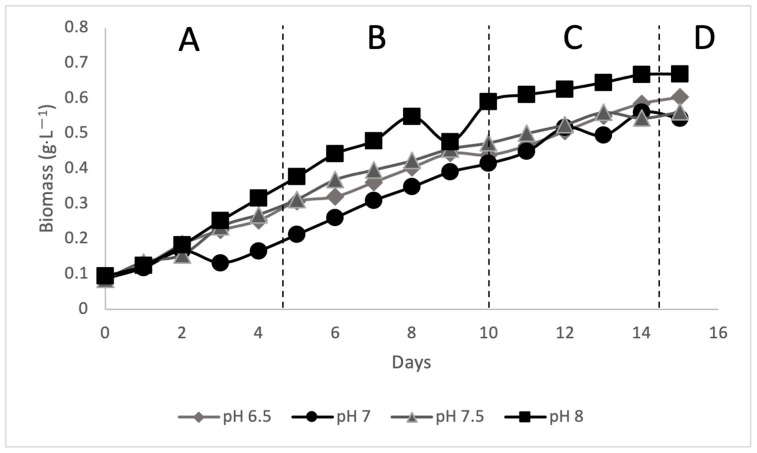
The growth of cell biomass via batch bioreactor.

**Figure 3 microorganisms-10-00880-f003:**
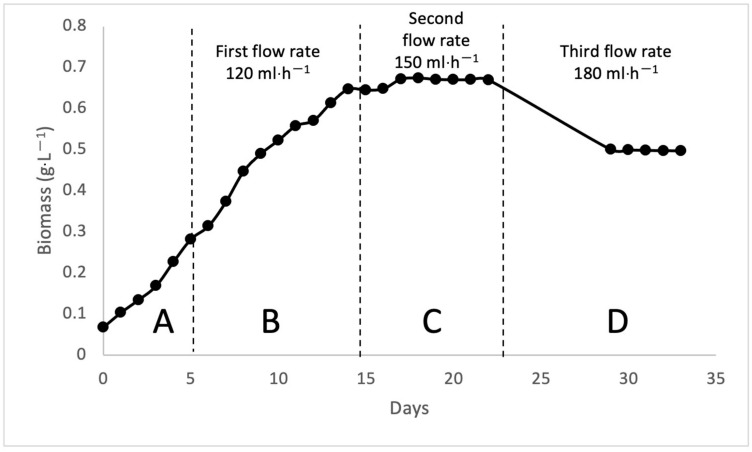
The growth of cell biomass for different pH via continuous bioreactor.

**Figure 4 microorganisms-10-00880-f004:**
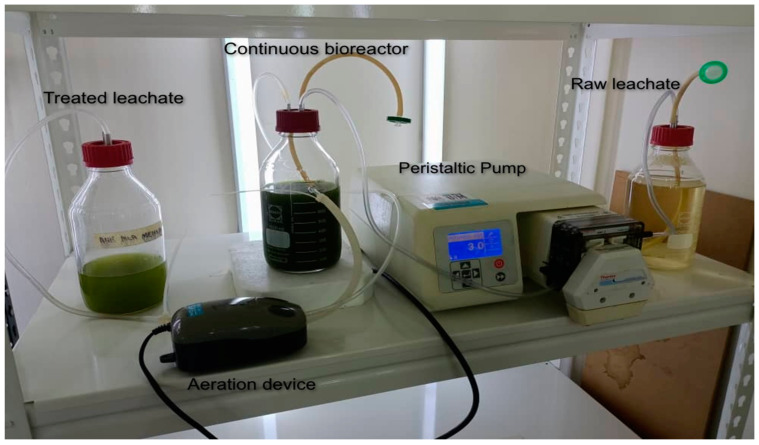
Laboratory photograph of the experiment.

**Figure 5 microorganisms-10-00880-f005:**
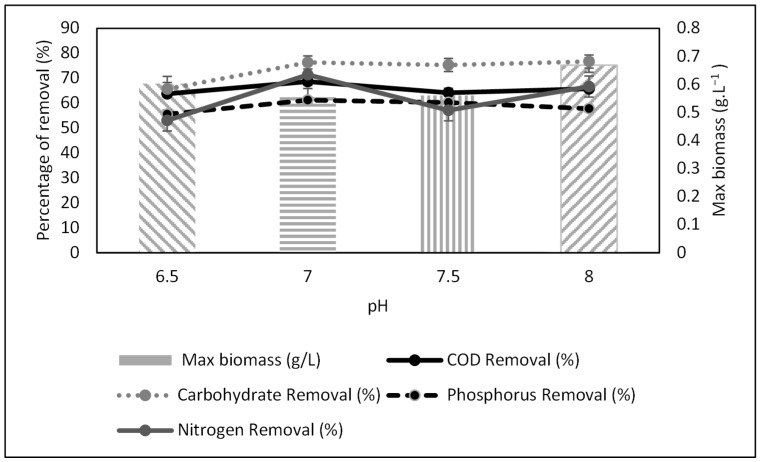
Physicochemical characteristics of treated raw leachate after batch cultivation.

**Figure 6 microorganisms-10-00880-f006:**
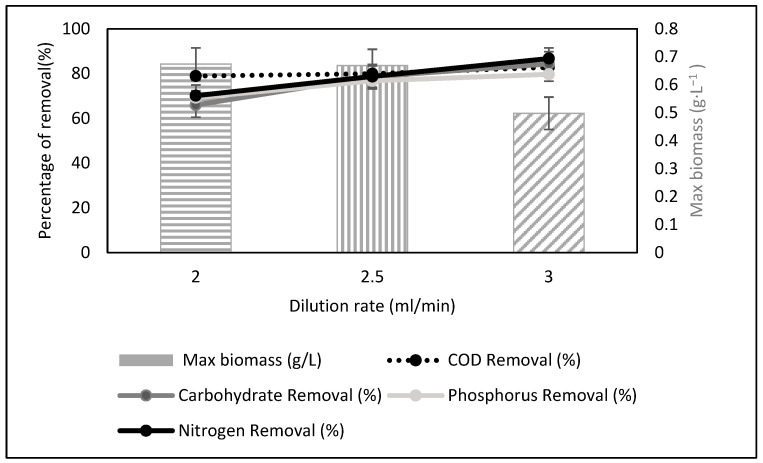
Physicochemical characteristics of treated raw leachate after continuous cultivation.

**Figure 7 microorganisms-10-00880-f007:**
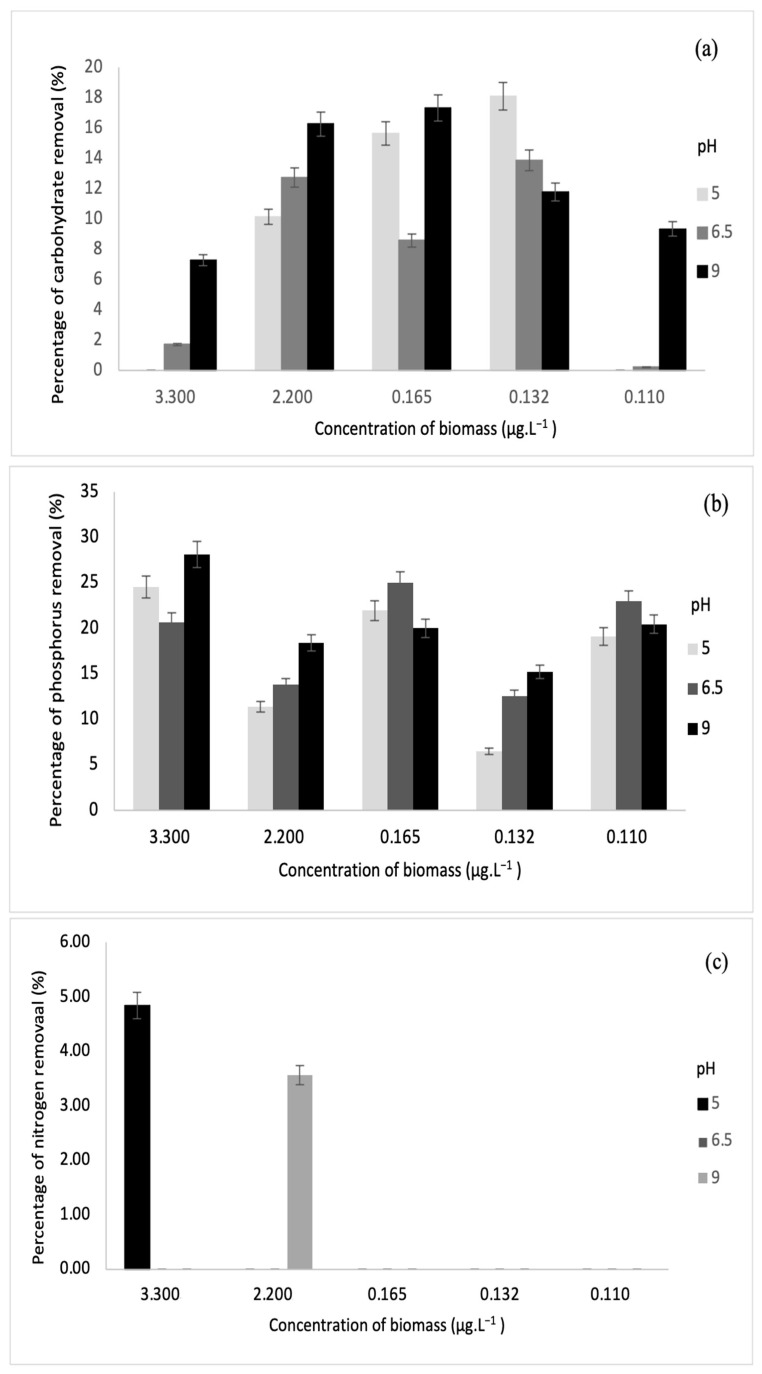
Relationships between biomass concentrations (with sonication) and (**a**) carbohydrate removal, (**b**) phosphorus removal, and (**c**) nitrogen removal.

**Figure 8 microorganisms-10-00880-f008:**
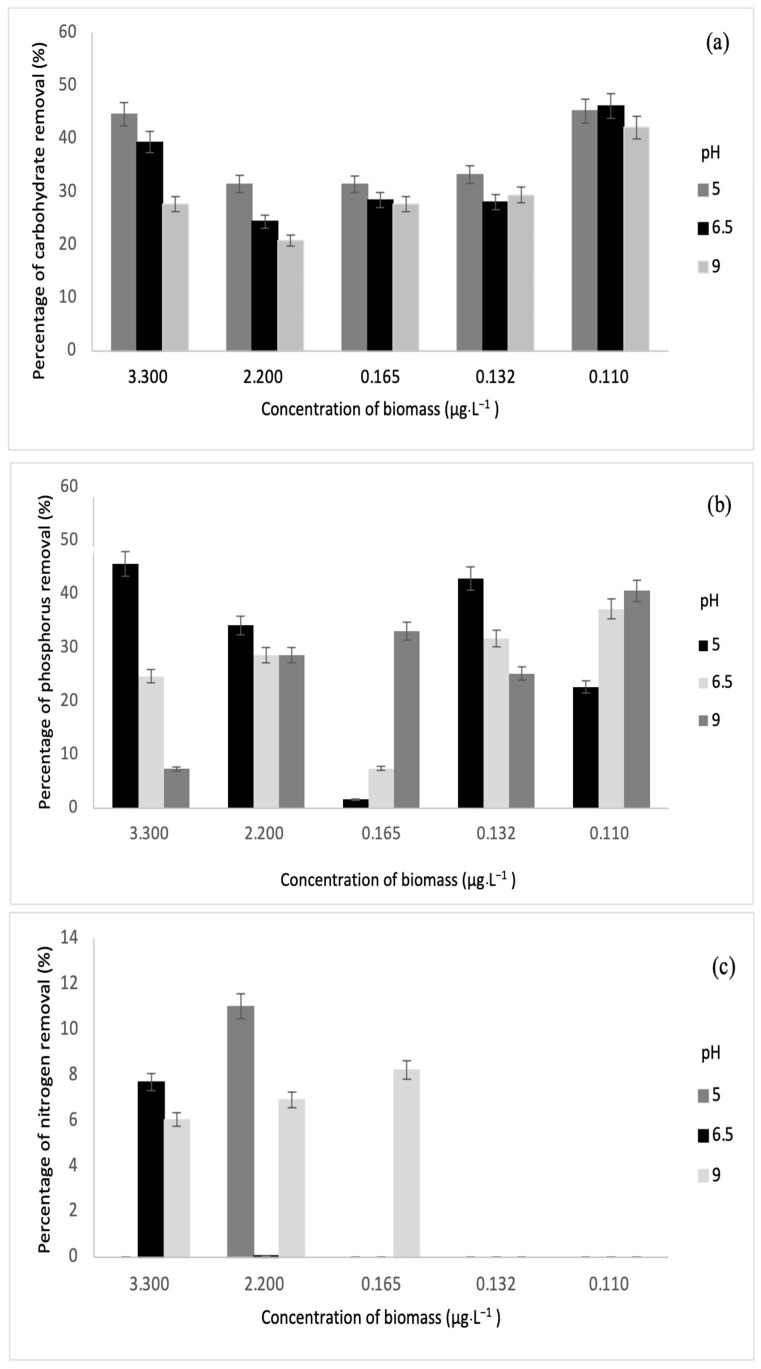
Relationships between biomass concentrations (without sonication) and (**a**) carbohydrate removal, (**b**) phosphorus removal, and (**c**) nitrogen removal.

**Figure 9 microorganisms-10-00880-f009:**
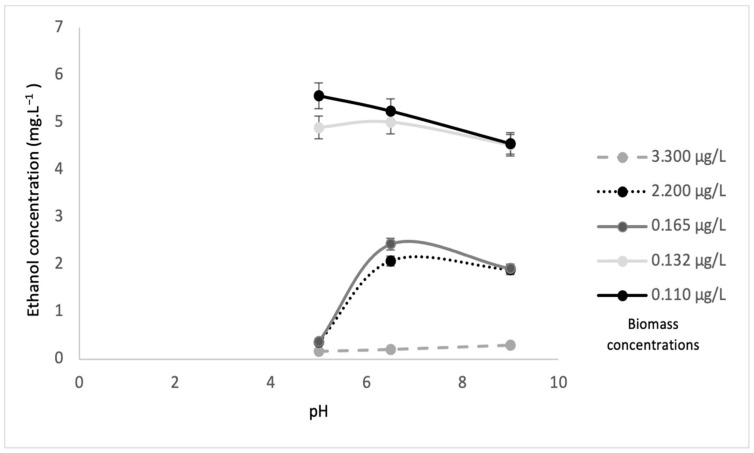
Ethanol concentration from *Scenedesmus* sp. biomass with sonication at different pH values.

**Figure 10 microorganisms-10-00880-f010:**
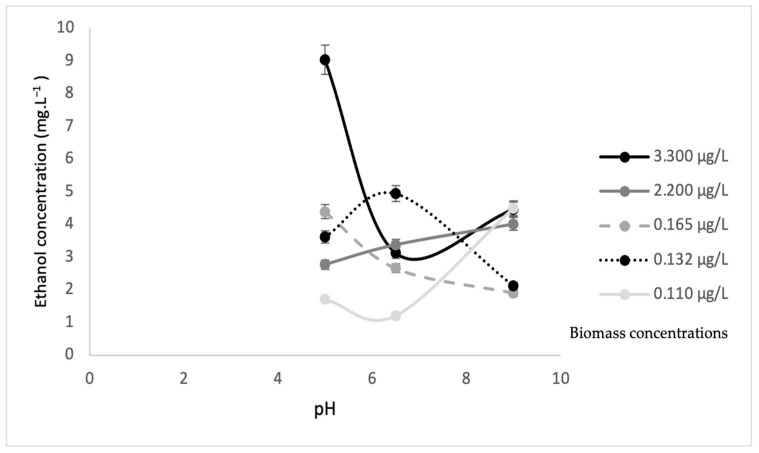
Ethanol concentration from *Scenedesmus* sp. biomass without sonication at different pH values.

**Table 1 microorganisms-10-00880-t001:** Characteristics of raw leachate used in this research.

Characteristics	Value
pH	5.250
Chemical oxygen demand, COD	6583 mg·L^−1^
Carbohydrate content	0.812 mg·L^−1^
Phosphorus content	4.340 mg·L^−1^
Total Kjeldahl nitrogen content (TKN)	245 mg·L^−1^

**Table 2 microorganisms-10-00880-t002:** Comparison of ethanol concentrations produced from various biomass concentrations (with and without sonication), pH values, and maximum biomass of *S. cerevisiae*.

Parameter		Ethanol Concentration (mg·L^−^^1^)		Maximum Biomass of *S. cerevisiae* (mg·L^−1^)	
Biomass Concentration (μg·L^−^^1^)	pH	With Sonication	Without Sonication	With Sonication	Without Sonication
	5	0.165	9.026	0.0361	0.0131
3.300	6.5	0.205	3.124	0.0387	0.0131
	9	0.291	4.450	0.0388	0.0290
	5	0.355	2.769	0.0304	0.0414
2.200	6.5	2.075	3.376	0.0333	0.0455
	9	1.885	4.016	0.0285	0.0526
	5	0.370	4.386	0.0451	0.0565
0.165	6.5	2.430	2,658	0.0272	0.0420
	9	1.909	1.901	0.0272	0.0370
	5	4.891	3,613	0.0571	0.0420
0.132	6.5	5.002	4.939	0.0572	0.0400
	9	4.521	2.114	0.0613	0.0395
	5	5.562	1.704	0.0631	0.0459
0.110	6.5	5.239	1.199	0.0653	0.0465
	9	4.552	4.489	0.0639	0.0467

## Data Availability

Not applicable.

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
