# Peer review of "Municipal Landfill Leachate Treatment and Sustainable Ethanol Production: A Biogreen Technology Approach"

_microorganisms, 2022, doi:10.3390/microorganisms10050880_

Round 1

Reviewer 1 Report

  1. References must be numbered in order of appearance in the text. Please cite the references according to Journal's guidelines.
  2. Lines 157-158

Please describe the isolation procedure and method for identification of the microalgae in detail. Also, provide crucial information on the cultivation of this microorganism, including temperature, time of cultivation or possible use of antibiotics for purifying microalga from bacteria.

  1. Lines 157-158: „The working volume for batch and continuous was 800 mL. The flow rates for fresh leachate feed into the bioreactor for a continuous system were 120, 150, and 180 mL/h“

It is well-known that the growth rate of microalgae is relatively low compared to yeast or bacteria. Therefore the dilution rate applied in the continuous cultivation of Scenedesmus seems to be too high. According to the calculation, dilution rates (0.15-0.225 h-1) are higher than the maximal values of the specific growth rate for most of the green microalgae. Please explain how you choose flow rates, i.e. dilution rates.

  1. Line 152-157

Since the Scenedesmus can grow mixotrophically and heterotrophically, please add text on cultivation conditions concerning the source of light. According to the scheme (Fig. 1), the bioreactor is made of transparent material that transmits light. Furthermore, the authors should describe the bioreactor (bubble-column, stirred tank reactor) and add important information on airflow rate and growth temperature…

  1. Lines 160-166

Please add text describing how inoculum was cultivated (temperature, growth medium, cultivation time).

  1. Lines 199-200: „Then, the substrate was inoculated with 10% (v/v) YEPG.“

The sentence is not clear. Please rewrite e.g. „Then, the growth medium was inoculated with 10 % (vol/vol) S. cerevisiae culture. „

  1. Lines 235-246

When presenting results, authors should use dilution rates instead of flow rates (Figure 3). In a steady-state, the specific growth rate is equal to the dilution rate and this parameter is an important characteristic that defines the growth kinetics of microorganisms. Furthermore, authors should compare the specific growth rates of Scenedesmus to other strains of the same genus or similar ones ( e.g. Chlorella) in the literature.

  1. Lines 222-223 :„The highest ethanol concentration of 9.020 mg/L was found in biomass concentration of 3.300 µg/L at pH 5 when fermentation occurred in samples without sonication“

The mass balance in the production of bioethanol is a big issue in the study. For example, in the experiment with the highest biomass concentration (3.300 µg/L) 9.020 mg/L of bioethanol is produced. If we assume that microalgal biomass used as a carbon source is glucose, the theoretical yield of ethanol (0.51 g ethanol/g glucose) would be 1.683 µg/L (i.e. 1.683 mg/L). The calculated ethanol concentration is significantly lower than obtained ethanol concentration. Result suggests that yeast used other carbon sources except the microalgal biomass for bioethanol production. The growth medium contained only microalgal biomass and inoculum grown on a complex growth medium according to the protocol.

  1. Figures 4 and 5

Correct the numbers on the x-axis (delete dot). E.g. instead of „3.300.“ it should be „3.300“.

  1. In this study, the microalgae biomass was used as a substrate for the growth of yeast and bioethanol production. Since the yeast can not digest the cell wall of the microalgae, the biomass of microalgae is usually pretreated (e.g.using acid) before fermentation to release components from the cell. However, in this study, ethanol concentrations were higher with microalgae biomass without ultrasound pretreatment than untreated biomass. Please explain this observation.

Author Response

Thank you so much for your revision time and your patience, I really appreciate your recommendation. all your comments already had been done.

Reviewer 2 Report

The authors have developed an interesting topic. I suggest the following corrections:

Methodology:
Provide the GPS coordinates of the landfill, in which year it was established (how long it has been in operation).
Add what types of waste are disposed of at the landfill (give statistical data for the last years, at least 5 years).
In what period (when) the leachate was collected.
Leachate changes its composition during the seasons and years of operation of the landfill. It would be useful to carry out sampling at different times of the year and compare the results.
Supplement the article with photographs of the experiment (real conditions and laboratory conditions).

Results:
Place the label (pH) correctly in the graph in Figure 4.
Fig. 4, 5, 6 and 7 show the values on the y-axis without decimal places.
Fig. 7 correct the label (Biomass - Biomass concentrations).
Check the quality and execution of all graphs in the paper.

Units should be given in the correct format (mg.L-1).

Author Response

(The authors gave the same response as above.)
